# A Randomized Controlled Trial of Guided Bone Regeneration for Peri-Implant Dehiscence Defects with Two Anorganic Bovine Bone Materials Covered by Titanium Meshes

**DOI:** 10.3390/ma15155294

**Published:** 2022-08-01

**Authors:** JaeHyung Lim, Sang Ho Jun, Marco Tallarico, Jun-Beom Park, Dae-Ho Park, Kyung-Gyun Hwang, Chang-Joo Park

**Affiliations:** 1Department of Oral and Maxillofacial Surgery, Korea University Ansan Hospital, Ansan-si 15355, Korea; surgidenta@gmail.com; 2Department of Oral and Maxillofacial Surgery, Korea University Anam Hospital, Seoul 02841, Korea; junsang@korea.ac.kr; 3Department of Medicine, Surgery, and Pharmacy, University of Sassari, 07100 Sassari, Italy; me@studiomarcotallarico.it; 4Department of Periodontics, College of Medicine, The Catholic University of Korea, Seoul 06591, Korea; jbassoonis@yahoo.co.kr; 5Division of Oral & Maxillofacial Surgery, Department of Dentistry, College of Medicine, Hanyang University, Seoul 04763, Korea; eogh392123@naver.com (D.-H.P.); hkg@hanyang.ac.kr (K.-G.H.)

**Keywords:** bone substitutes, clinical research, clinical trials, CT imaging, guided tissue regeneration, bone regeneration, surgical techniques

## Abstract

The aim of this study is to compare two low-temperature sintered anorganic bovine bone materials (ABBMs), Bio-Oss (Geistlich, Wolhusen, Switzerland) and A-Oss (Osstem, Seoul, Korea), for GBR in dehiscence defects. A single implant was placed simultaneously with GBR in the buccal or bucco-proximal osseous defect by double-layering of inner allograft and outer ABBM, covered by a preformed ultrafine titanium mesh and an absorbable collagen membrane. Grafted volume changes were evaluated by cone-beam computed tomography, taken preoperatively (T0), immediately after implant surgery (T1), after re-entry surgery (T2), and after delivery of the final restoration (T3). The density of the regenerated bone was assessed by measuring the probing depth on the buccal mid-center of the mesh after removing the mesh at T2. Postoperative sequelae were also recorded. Grafted volume shrinkage of 46.0% (0.78 ± 0.37 cc) and 40.8% (0.79 ± 0.33 cc) in the Bio-Oss group (8 patients) and A-Oss group (8 patients), respectively, was observed at T3 (*p* < 0.001). There were no significant differences in grafted volume changes according to time periods or bone density between the two groups. Despite postoperative mesh exposure (3 patients), premature removal of these exposed meshes and additional grafting was not necessary, and all implants were functional over the 1-year follow-up period. Both ABBMs with titanium meshes showed no significant difference in the quantity and density of the regenerated bone after GBR for peri-implant defects.

## 1. Introduction

Innovative devices and technologies to reduce morbidity, biological, and surgical times are an intense research topic in implant dentistry [1], and guided bone regeneration (GBR) is a surgical procedure of bone augmentation for dental implants, using various bone grafting materials with concurrent barrier membranes [2]. Anorganic bovine bone materials (ABBMs) have been widely used as an osteoconductive material for sinus lift surgeries, ridge preservation, implant surgery immediately after tooth extraction, and GBR [3,4,5,6,7]. The unique advantage of ABBM is its own property of volume maintenance with slow substitution by the new bone compared to autogenous and allogeneic bone grafting material. There are many other types of ABBMs that are sintered at high temperature [8] or are porcine- and equine-based [9]; however, bovine-based and low-temperature-sintered ABBMs appear to dominate the dental market [10]. Historically, the Bio-Oss (Geistlich, Wolhusen, Switzerland) has long been considered the gold standard due to strong scientific evidence; nevertheless, A-Oss (Osstem, Seoul, Korea) is an ABBM produced by chemical treatment with the aromatic and strong alkali solvent and finally by a low-temperature annealing process below 400 °C with an extremely low heating rate (<0.3 °C per minute) [11]. The physicochemical characterization of A-Oss, which could be substantially equivalent to that of Bio-Oss in view of both higher porosity and lower crystallinity, is closely associated with less degradation and remodeling activity compared to other xenografts (Figure 1) [11,12].

In the field of oral and maxillofacial surgery, the titanium mesh has been widely adopted for cases requiring a large amount of bone reconstruction due to its inherent biocompatibility and rigidity as a light metal [13,14,15]. Particularly, the recent introduction of a preformed ultrafine titanium mesh, which is directly connected and fixed upon the implant, has increased its popularity in GBR for peri-implant osseous defects [16,17]. Resistant to collapse during the entire healing period of GBR, this ultrafine titanium mesh has multiple pores with varying sizes for optimal GBR results [18].

The purpose of this randomized controlled clinical trial was to compare two low-temperature-sintered ABBMs, Bio-Oss and A-Oss, which are used in GBR for the reconstruction of peri-implant dehiscence defect, in combination with the ultrafine titanium mesh. To our knowledge, this is the first study to measure and compare three-dimensional volumetric changes of ABBMs for GBR purposes. The null hypothesis was that there was no difference in changes in the grafted volume (bone quantity), density (bone quality) of the regenerated bone, and postoperative sequelae between the groups.

## 2. Materials and Methods

This study was designed as a randomized controlled trial of parallel group design and was conducted at the Division of Oral and Maxillofacial Surgery, Department of Dentistry, Hanyang University Hospital, between June 2018 and July 2019. The study was approved by the Institutional Review Board of Hanyang University Hospital (IRB No. 2018-03-006) and registered at the WHO international clinical trials registry platform (KCT0004906). It was conducted in accordance with the Declarations of Helsinki and internationally accepted guidelines for RCTs, including the CONSORT statement (accessed on 2 March 2018). All the surgical and prosthetic procedures were performed by one expert implantologist (C.-J.P.).

### 2.1. Patient Selection

After preoperative clinical and radiologic examinations, including cone-beam computed tomography (CBCT), any healthy patient aged 18 years or older, who required a single implant placement in the healed alveolar ridge with GBR for the peri-implant dehiscence defect when the implant was simulated in a prosthetically driven position (OneGuide, Osstem, Seoul, Korea), was included in this study after providing informed consent. Particularly, the osseous defects were to be limited to the buccal and/or proximal aspect only and did not involve the adjacent tooth surface or lingual aspect. The exclusion criteria were requiring more than two consecutive implants and having systemic or local contraindications for implant placement, including a history of uncontrolled metabolic disorders, smoking, bruxism, or uncontrolled periodontal disease. Enrolled patients were informed about the surgical procedures, materials to be used, benefits, and the potential risks and complications of this clinical study, and written informed consent was obtained.

### 2.2. Surgical Procedure

All procedures were performed under local anesthesia and moderate intravenous sedation. A full-thickness flap was raised, and all granulation tissue was thoroughly removed. Initial drilling with cortical bone marking was performed by a guide drill. Sequential taper drills were used to prepare the implant site (OneGuide kit or 122 Taper kit, Osstem), and special emphasis was laid on obtaining a correct three-dimensional position of the implant. A dental implant (TS III SOI, Osstem, or T01 SA, Toplan, Seoul, Korea) was installed 1 mm sub-crestally to the lingual or palatal alveolar crest, and a dehiscence defect occurred around the implant top, exposing the buccal and/or proximal implant threads.

Defect size (mm) consisted of the defect height, which was measured from the implant top to the first bone-to-implant contact, and defect width, which was measured from the mesial to the distal bone crest at the level of the implant top (Figure 2a,b). The preformed ultrafine titanium mesh (OssBuilder, Osstem), which was the most appropriate to the measured defect size, was selected from various line-ups of type I for buccal dehiscence defect and type II for bucco-proximal combined dehiscence defect (Figure 3). Minor trimming and contouring of the titanium mesh were conducted for better containment of the grafting material if necessary. Multiple bone marrow openings were performed to induce the osteogenic cells for faster and better bone regeneration.

According to the randomization envelope, patients were assigned to two groups: Bio-Oss group and A-Oss group. The exposed implant threads were covered by a freeze-dried bone allograft (FDBA; SureOss, HansBiomed, Seoul, Korea) and overlaid by an ABBM, Bio-Oss, or A-Oss, in a 1:1 volume ratio for contour maintenance. The anchoring part, which connects the implant and preformed ultrafine titanium mesh, was hand tightened on the implant, and the selected ultrafine titanium mesh was applied on this anchor to contain the whole grafting material (Figure 3). If the implant was inserted with a seating torque ≤30 Ncm or an implant stability quotient value ≤70, the cover cap was connected to the titanium mesh according to a submerged approach. If not, the healing cap was immediately connected according to a non-submerged approach. An absorbable collagen membrane (OssGuide, SK Bioland, Cheonan, Korea) or OssMem hard (Osstem) was adapted upon the titanium mesh to minimize the risk of thinning of the overlying gingiva resulting in the early exposure of the titanium mesh. The periosteum of the full-thickness flap was released for tension-free primary wound closure. Patients received analgesics and antibiotics for 10 days to control postsurgical pain and infection. Chlorhexidine mouth rinses were also prescribed twice daily. Sutures were removed 10 days after the surgery.

At 8 months postoperatively, re-entry surgery was performed using a similar flap design under local anesthesia. After the titanium mesh was removed, bone density was evaluated by the probing depth, which was measured in the regenerated bone beneath the mid-center of the titanium mesh, using controlled force, 0.25 N (Figure 2c). The cover cap or healing cap was replaced by a healing abutment, and the flaps were adjusted and sutured. Sutures were removed 10 days after the surgery. After the re-entry surgery, the soft tissue around the healing abutment was allowed to heal before impression-taking. A single temporary restoration was incorporated with the titanium custom abutment for progressive loading, and the final restoration was delivered at 12 months postoperatively. Any postoperative sequelae in GBR procedures were recorded according to the previous classification of GBR complications [19] and treated by the same surgeon (C.-J.P.).

### 2.3. Evaluation of Grafted Volume

Changes in the augmented volume of the grafting material were evaluated by comparing the CBCT data taken preoperatively (T0), immediately after the implant surgery and GBR (T1), after the re-entry surgery (T2), and after the delivery of the final restoration (T3). Exposure parameters for CBCT scanning (field of vision 35 × 50 mm; voxel size 0.2 mm; time 10.8 s; kV 75; mA 10; dose area product 240 mGy·cm^2^) were set as low as reasonably achievable (ALARA). Using Aquarium iNtuition software (TeraRecon, Durham, NC, USA), the implant site was reconstructed into a three-dimensional image and segmented as a region of interest (ROI) using nearby anatomic structures, such as adjacent teeth, sinus floor, and inferior alveolar canal. Compartments, which are radiologically uniform in density and size, such as implant body and healing abutment, were subtracted, and volume (cc) in ROI was calculated according to time periods (Figure 4) by an independent blinded examiner (K.-G.H.). Since this software is widely used in the medical field to precisely detect the occluded portion of major vessels [20] and accurately measure the volume of the organ for transplantation [21], it was applied for this study to measure the ROI volume to complement CT technology.

### 2.4. Statistical Analysis

The sample size was determined to satisfy 90% power of the test at a minimum at α = 0.05 according to the equal variance assumption based on the within-subject and between-subject means from the previous randomized controlled study comparing Bio-Oss and the calcium phosphate-coated ABBM in sinus augmentation [22]. Based on these results, the minimum sample size was estimated to be 9 subjects in each group, considering a possible attrition rate of 10% during the study period.

The mixed-effect model was used to examine the differences in grafted volume changes between Bio-Oss and A-Oss groups with the adjustment for covariates, where the grafted volume changes were measured at three consecutive time periods as well as at baseline (T0). The significance of the covariate effects and the predicted grafted volume changes were tested at α = 0.05. In addition, descriptive statistics about the general information of the patients at baseline were summarized with the number of cases and percentage for categorical variables and with the mean ± standard deviation for continuous variables. All statistical analyses were performed using SAS, version 9.4 (SAS Inc., Cary, NC, USA). Statistical significance was set as *p* < 0.05.

## 3. Results

A total of 18 patients were consecutively enrolled, and two patients dropped out (one in each group) due to loss of follow-up. Finally, data from 16 patients (7 men and 9 women) with a mean age of 54.3 ± 11.7 years (range from 28 to 72 years) were collected and evaluated in this trial (Table 1). The demographic comparison between the two groups is shown in Table 2. There was a significant difference between the Bio-Oss and A-Oss groups only in terms of the implant length (*p* < 0.01).

In both groups, the grafted volume, which peaked at 1.70 ± 0.50 cc in the Bio-Oss group and 1.94 ± 0.26 cc in the A-Oss group at T1, continued to shrink up to T3. The grafted volumes reduced to 1.17 ± 0.38 cc in the Bio-Oss group and 1.46 ± 0.14 cc in the A-Oss group at T2 and 0.92 ± 0.37 cc in the Bio-Oss group and 1.15 ± 0.16 cc in the A-Oss group at T3; there was a significant difference in the grafted volumes between time periods (*p* < 0.001, Table 3). However, there was no significant difference between the two groups at all time periods (*p* > 0.05, Figure 5). Furthermore, there was no significant difference in the densities of the regenerated bone at T3 between the groups (*p* > 0.05).

In mixed-effect model analysis, T2 and T3 as time periods (*p* < 0.001), older age (*p* < 0.01), and the existence of sequelae (*p* < 0.05) indicated significant decrements in the grafted volume; conversely, wider defect (*p* < 0.01) and maxilla as the surgical site (*p* < 0.05) resulted in lesser decrements in the grafted volume (Table 4). Postoperative sequelae were noted in three patients, with two patients (Class I and II at crestal suturing sites) in the Bio-Oss group and one (Class I at bucco-mesial edge of mesh) in the A-Oss group. All postoperative sequelae occurred in submerged cases with cover caps, and there was no significant difference between the two groups. Patients were instructed to brush the mesh carefully and gently with a soft toothbrush soaked with 1% chlorhexidine gel twice daily. Despite no re-epithelization covering the exposed area of the mesh, premature removal of these exposed meshes was not necessary, and re-exposure of implant threads and severe loss of the grafting material with suppuration were not observed during the entire healing period; no additional grafting was necessary at T2.

## 4. Discussion

This randomized controlled trial was designed to evaluate the potential of A-Oss to maintain the bone quantity and quality of the regenerated bone by GBR as a low-temperature-sintered ABBM compared to Bio-Oss, and the null hypothesis of no difference was accepted. Despite the use of volume-stable ABBMs, a total of 46.0% (0.78 ± 0.37 cc) and 40.8% (0.79 ± 0.33 cc) of grafted volumes were lost during the process of substitution by new bone in the Bio-Oss and A-Oss groups, respectively. A possible explanation for greater resorption beyond expectation is that the grafted volume measured at T1 was clearly outlined by thin radiopaque titanium mesh on CBCT scans, whereas the indistinct radiologic periphery between grafting materials and soft tissue resulted in underestimation of the grafted volume after the removal of the titanium mesh at both T2 and T3. Compared to 17.3~23.9% of grafted volume loss until postoperative 6 months after sinus lifting surgery [22,23], the shrinkage of grafted volume after GBR was assumed to be greater because the overall pressure of the overlying soft tissue may be higher than the intrasinus pneumatic pressure. Only linear measurements were carried out to assess the thickness of the augmented region by GBR in limited CBCT scans, and our results are not surprising, considering that the linear reduction ranged from 20.2% to 42.8% at the implant shoulder after grafting of Bio-Oss [24,25]. However, we should admit that there was a constant risk of measuring errors resulting from metal artifacts because various metallic components, such as an implant body and a titanium mesh, existed simultaneously in CBCT.

Focusing on the grafted volume change, the effects of the existence of sequelae and the surgical site showed statistically significant differences (*p* < 0.05). In other words, if the GBR had postoperative complications or the surgical site was posterior (vs. anterior, *p* = 0.0583) and not the maxilla (vs. mandible), more resorption of the grafted volume was likely to occur. The mandibular posterior area is generally composed of hard bone with a thicker cortex, and the result of bone augmentation is less predictable even if combined with extensive bone marrow openings [26]. In addition, the effects of age and defect width also showed significant differences (*p* < 0.01), meaning that older patients and more narrow dehiscence defects will be likely to show more resorption of the grafted volume. Older patients tended to present a higher risk of implant failure with lower potentials of bone regeneration after GBR [27], and there was less resorption at the GBR site as more bone grafting materials were packed in wider defects and protected securely by a titanium mesh maximizing the volume-stable property of ABBMs. Preformed ultrafine titanium mesh was suitable for the ideal bone contour, particularly at the level of the implant top, preventing the collapse of the graft [16,17,18]. Supposing that peri-implant defect is located within the original bone housing, fenestration-type defects have shown more bone fill and fewer complications than dehiscence-type [28]. Thus, we cannot find the significant influence of defect height on grafted volume change.

In contrast to the ABBM for volume preservation, allogeneic bone grafting materials depicted robust bone formation with islands of new bone that might be interpreted as evidence of bone induction and appeared to be in a more active state of turnover and replacement [29]. A layering technique using different grafting materials is commonly applied to augment dehisced or deficient alveolar bone around dental implants [30]. Generally, autogenous bone was used as an inner layer in close contact with the implant; however, allogeneic bone was also used when autogenous bone harvest volume was not adequate [29,31]. In this study, an outer layer of ABBM was placed on the allograft to preserve and maintain the augmented bone, and a third layer of absorbable collagen membrane was placed to prevent the soft tissue and non-osteogenic cell invasion into the grafted site [4,30]. In view of the prerequisite of a barrier membrane, titanium mesh was used for space-making ability and stabilization of blood clots, and the overlying absorbable collagen membrane was used for cell occlusiveness to prevent the premature thinning of the overlying gingiva by separating the compartments completely for osseous and epithelial regeneration [15]. As a non-absorbable barrier membrane, the removal of titanium mesh is unavoidable; however, the re-entry surgery to remove it is a perfect chance to evaluate the quantity and quality of the regenerated bone for the determination of the success of GBR. Commonly, while bone quantity can be assessed radiologically by 3-dimensional reconstruction of CBCT scans at the GBR site [32], bone quality has been evaluated histologically or histomorphometrically by obtaining and analyzing the bone core biopsy from the regenerated ridge [33]. Nevertheless, in our study, the bone quality was grossly assessed by evaluating the corticalization of the GBR site using probing depth during the re-entry because it was technically hard or frequently impossible to obtain the bone sample from the reconstructed peri-implant site. After removal of the titanium mesh, closer inspection of the healed site revealed regenerated hard tissue superficially covered by a thin layer of soft tissue that was 1–2 mm thick [13]. This layer was described as “pseudo-periosteum” and the clinical significance of this histologically connective and granulation tissue layer is unknown [34]; however, this thin layer seems stable in dimension, and its removal was not indicated during the re-entry [16]. In our study, probing depths of 2.8 ± 0.7 mm and 2.6 ± 0.8 mm in Bio-Oss and A-Oss groups, respectively, could be the thickness of immature bone, including the “pseudo-periosteum”, during the process of corticalization beneath the titanium mesh. This approach to evaluating the quality of the regenerated bone is presumed to be the main limitation of our study and further histologic and histomorphometric investigation will be needed to analyse the regenerated bone in detail.

Even though no re-epithelization occurred on small (2 patients) and large (1 patient) exposures of titanium mesh without purulent exudate during the healing period, leak of grafting material was not observed, and no premature removal of the exposed titanium mesh and no supplemental grafting were indicated. However, the probing depth increased up to 3~4 mm due to the compensatory presence of thicker “pseudo-periosteum”, which was involved in keeping the grafting material securely contained in cases of exposed titanium mesh. A previous study also confirmed that exposure to the titanium mesh had no negative influence on the clinical outcome of the augmentation procedure and the success of the bone grafting procedure [35]. Especially, all postoperative sequelae occurred in cases where titanium mesh was fixed and stabilized with cover caps, though it was not statistically significant. As cover caps were connected to the implant with low primary stability in our study, more tension resulting from less redundant soft tissue might be attributed to a higher incidence of postoperative wound dehiscence in the submerged approach compared to the non-submerged approach. There was no significant difference in the postoperative sequelae between the Bio-Oss and A-Oss groups.

## 5. Conclusions

Within the limitations of this study, it can be concluded that

A peri-implant dehiscence defect was successfully reconstructed by GBR with double-layering of allograft and ABBM, which were covered by a preformed ultrafine titanium mesh and an absorbable collagen membrane;Despite the volume maintenance effect of ABBMs, approximately 27.8% of grafted volume resorption was noted at T2, and there was no significant difference between the Bio-Oss and A-Oss groups, even in the quality of the regenerated bone;The grafted volume loss (approximately 43.2%) continued up to T3 and did not differ between the Bio-Oss and A-Oss groups.

## Figures and Tables

**Figure 1 materials-15-05294-f001:**
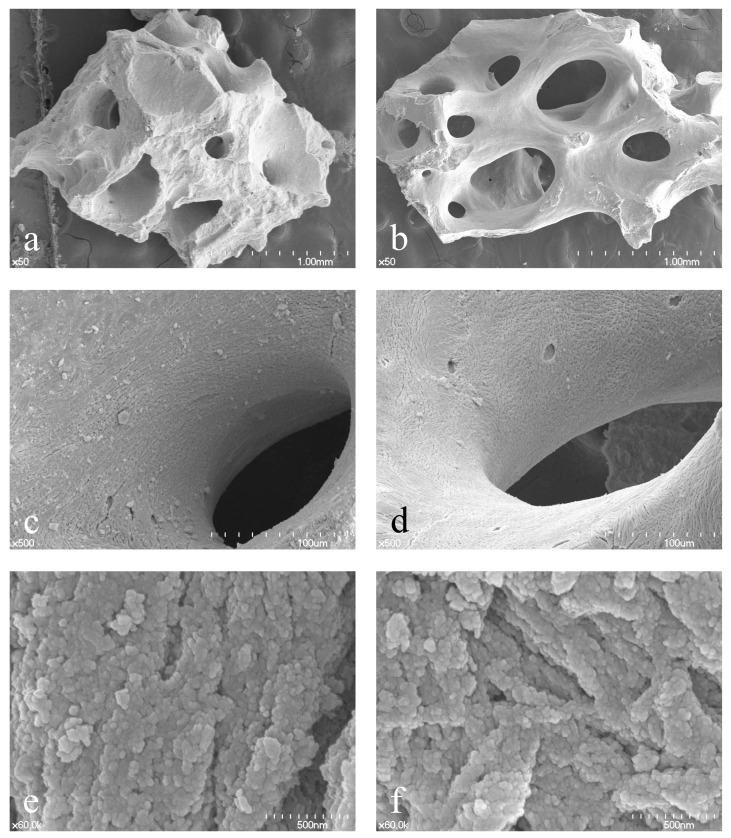
Morphological comparisons of Bio-Oss and A-Oss by scanning electron microscope. Bio-Oss at ×50 (**a**), ×500 (**c**), and ×60,000 magnification (**e**); A-Oss at ×50 (**b**), ×500 (**d**), and ×60,000 magnification (**f**).

**Figure 2 materials-15-05294-f002:**
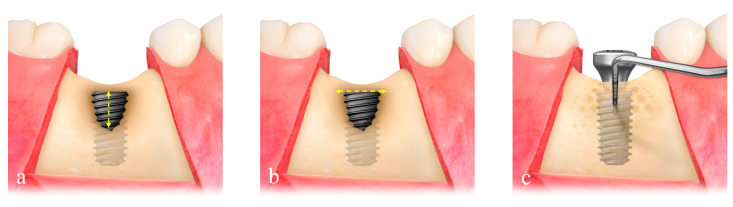
Measurement of the defect size composed of defect height (**a**) and defect width (**b**), respectively, and evaluation of the bone density of the regenerated bone by probing depth (**c**).

**Figure 3 materials-15-05294-f003:**
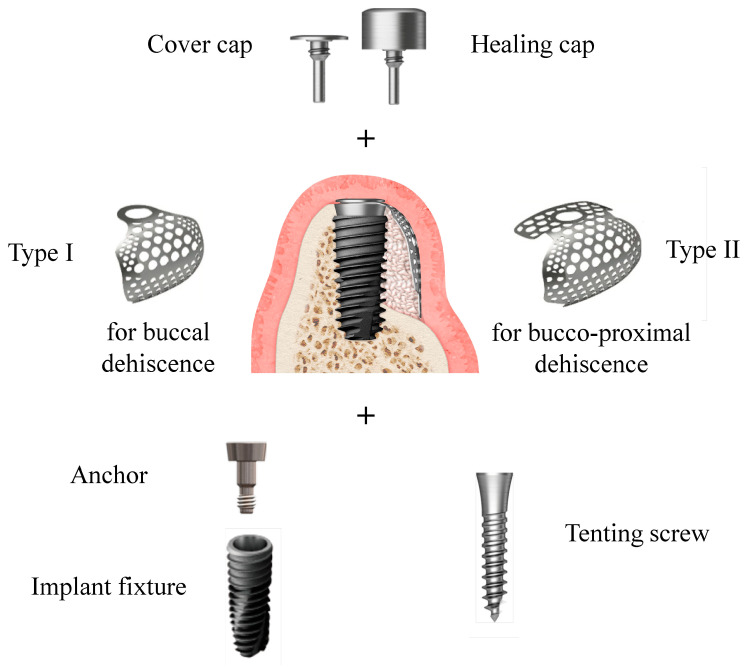
Components of preformed ultrafine titanium mesh used in this study for GBR on the peri-implant dehiscence defect.

**Figure 4 materials-15-05294-f004:**
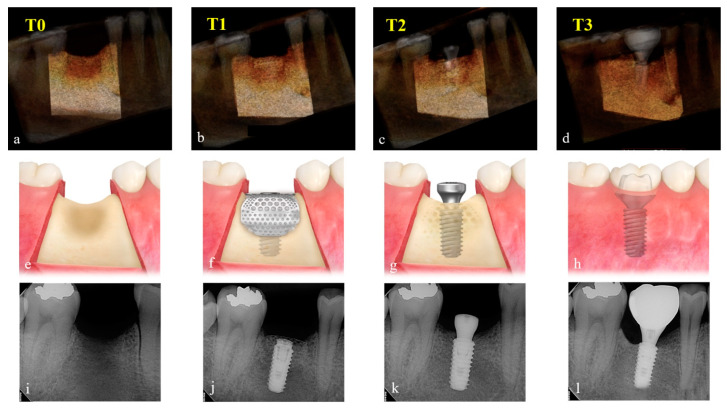
CBCT scans, illustrations, and periapical radiographs were shown according to time periods, and grafted volumes were calculated from three-dimensionally reconstructed CBCT scans by Aquarium iNtuition software, version 4.4.12 (TeraRecon, Durham, NC, USA). T0: Preoperatively (**a**,**e**,**i**), T1: Immediately after the implant surgery and GBR (**b**,**f**,**j**), T2: After the re-entry surgery (**c**,**g**,**k**), T3: After the delivery of the final restoration (**d**,**h**,**l**).

**Figure 5 materials-15-05294-f005:**
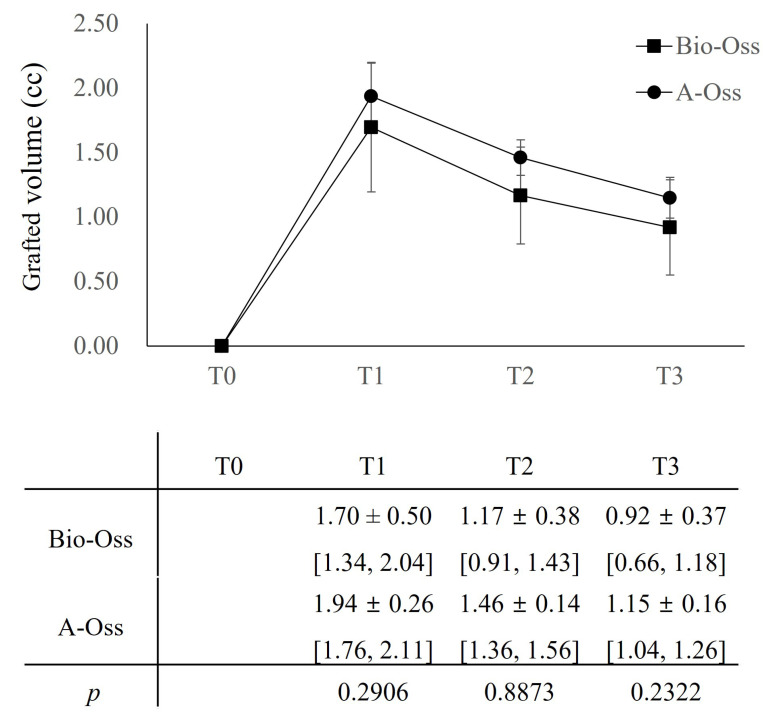
Comparison of the grafted volumes between Bio-Oss group and A-Oss group according to time periods by *ad hoc* test of Bonferroni correction after repeated analysis of variance. Mean ± SD (mm) with a 95% confidence interval (CI). T0: Preoperatively, T1: Immediately after the implant surgery and GBR, T2: After the re-entry surgery, T3: After the delivery of the final restoration.

**Table 1 materials-15-05294-t001:** Patient characteristics.

ABBMs	Patient			Site (FDI)	Implant Body (mm)		Defect Size (mm)		Grafted Volume (cc)				Probing Depth (mm)	Titanium Mesh		Collagen Membrane	Sequelae
		Sex (M:F)	Age (Year)		Width	Length	Width	Length	T0	T1	T2	T3		C:H	Type 1:2	OG:OM	Class I:II
Bio-Oss group	1	F	63	36	5	7	6	3		1.14	0.91	0.80	3.5	H	2	M	
	2	Dropped															
	3	M	56	16	4.5	8.5	5	4.5		1.53	1.11	0.89	2	H	1	G	
	4	M	60	23	4	8.5	6.5	4.5		1.91	1.40	1.10	3	C	2	G	I
	5	F	54	42	4	8.5	4	4	0.00	1.16	0.83	0.60	2.5	H	1	M	
	6	F	55	11	4	10	4	4		2.41	1.32	0.85	3	H	1	M	
	7	M	51	16	4.5	8.5	4.5	3		1.33	0.91	0.67	2	H	1	G	
	8	F	28	12	4	8.5	5.5.	3		2.37	1.94	1.75	2	C	2	M	
	9	F	63	46	5	7	7	4		1.72	0.91	0.70	4	C	2	M	II
A-Oss group	1	M	30	46	5	10	5	3		1.53	1.29	1.17	3	H	1	G	
	2	F	54	16	4.5	8.5	6.5	5		2.04	1.51	1.20	3.5	H	2	G	
	3	M	50	42	4	8.5	6	3.5		1.98	1.67	1.25	2	C	2	M	
	4	Dropped															
	5	M	53	35	4	10	6	6	0.00	1.73	1.39	1.18	2	H	2	M	
	6	F	64	15	4	8.5	6	7		2.12	1.60	1.37	2	H	2	G	
	7	F	65	25	4	10	6.5	2.5		1.73	1.29	1.18	1.5	C	2	M	
	8	F	72	21	4	8.5	4	5.5		2.03	1.42	0.87	3	H	1	M	
	9	M	50	33	4	8.5	6	4		2.33	1.51	0.97	4	C	2	M	I

ABBMs: anorganic bovine bone materials, C: cover cap, H: healing cap, OG: OssGuide, OM: OssMem hard. Type I: preformed titanium mesh for buccal dehiscence defect, Type II: preformed titanium mesh for bucco-proximal combined dehiscence defect. Class I: small membrane exposure (≤3 mm) without purulent exudate, Class II: large membrane exposure (>3 mm) without purulent exudate. T0: Preoperatively, T1: Immediately after the implant surgery and GBR, T2: After the re-entry surgery, T3: After the delivery of the final restoration.

**Table 2 materials-15-05294-t002:** Demographic comparison of the patients between Bio-Oss and A-Oss groups.

Variables		Bio-Oss	A-Oss	*p*-Value
Sex	(Male)	3 (37.5)	4 (50.0)	0.6143
	(Female)	5 (62.5)	4 (50.0)	
Age	(Years)	53.8 ± 10.8	54.8 ± 12.2	0.9258
Surgical site	(Maxilla)	5 (62.5)	4 (50.0)	0.6143
	(Mandible)	3 (37.5)	4 (50.0)	
	(Anterior)	4 (50.0)	3 (37.5)	0.6143
	(Posterior)	4 (50.0)	5 (62.5)	
Defect	(Width)	5.3 ± 1.1	5.8 ± 0.8	0.1542
	(Height)	3.8 ± 0.6	4.6 ± 1.5	0.0892
Implant (mm)	(Width)	4.4 ± 0.4	4.2 ± 0.4	0.0850
	(Length)	8.3 ± 0.9	9.1 ± 0.7	0.0050 *
Probing depth (mm)	(mm)	2.8 ± 0.7	2.6 ± 0.8	0.5001
Titanium mesh	(Cover cap)	3 (37.5)	4 (50.0)	0.6143
	(Healing cap)	5 (62.5)	4 (50.0)	
	Type I (Buccal only)	4 (50.0)	2 (25.0)	0.2369
	Type II (Bucco-proximal)	4 (50.0)	6 (75.0)	
Sequelae	(Yes)	2 (25.0)	1 (12.5)	0.4103

The number of cases (percentage) was summarized for categorical variables and compared by Fisher’s exact test. Mean ± SD was summarized for continuous variables and * *p* < 0.01 by Wilcoxon’s test.

**Table 3 materials-15-05294-t003:** Comparison of grafted volumes between time periods.

	Estimates	*T* Value	*p*-Value
T2—T3	−0.2244	−5.02	0.0007 *
T2—T1	−2.3194	−15.62	<0.0001 *
T3—T1	−2.0950	−16.87	<0.0001 *

* *p* < 0.001 by comparison of least squares means (LSM) in the mixed-effect model at α = 0.05. T1: Immediately after the implant surgery and GBR, T2: After the re-entry surgery, T3: After the delivery of the final restoration.

**Table 4 materials-15-05294-t004:** Coefficients estimates for grafted volume changes by mixed-effect model.

Effects		Estimates	*T* Value	*p*-Value
Group		−0.2112	−1.12	0.2906
Time	(T2)	−2.4125	−11.49	<0.0001 ***
	(T3)	−2.2475	−12.80	<0.0001 ***
Group × Time †	(T2)	0.1863	0.63	0.5460
	(T3)	0.3050	1.23	0.2505
Age		−0.0045	−4.17	0.0024 **
Sequelae	(Yes)	−0.1184	−2.79	0.0210 *
Defect	(Width)	0.0796	4.58	0.0013 **
Surgical site	(Posterior)	−0.0624	−2.17	0.0583
	(Maxilla)	0.0684	2.77	0.0218 *

*** *p* < 0.001; ** *p* < 0.01; * *p* < 0.05 by the mixed-effect model at α = 0.05. † Analysis of the interaction effects between the group and time on the assumption that the grafted volume changes of two groups may interact with the observed time points. T2: Volume measured after the re-entry surgery, T3: Volume measured after the delivery of the final restoration. Effects of sex (male), titanium mesh (cover cap), titanium mesh (buccal only), and probing depth were not significant as *p* values were 0.8457, 0.0638, 0.2369, and 0.4256, respectively. Nor effects of implant (width), implant (length), and defect (height) were significant as *p* values were 0.4215, 0.0604, and 0.0969, respectively.

## Data Availability

All data are available upon request of the authors.

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
