# Peer review of "A Randomized Controlled Trial of Guided Bone Regeneration for Peri-Implant Dehiscence Defects with Two Anorganic Bovine Bone Materials Covered by Titanium Meshes"

_materials, 2022, doi:10.3390/ma15155294_

Round 1

Reviewer 1 Report

In this manuscript, the authors compared two low temperature-sintered ABBMs, Bio-Oss and A-Oss, which are used in GBR for the reconstruction of peri-implant dehiscence defect, in combination with the ultrafine titanium mesh. I would like to put forward the following questions and comments:

1. There are only 7 patients in each group, the sample size is comparatively small and may affects an RCT's reliability.

2. The implants and GBR in this experiment were placed in different sites, including both posterior and anterior areas on mandibular and maxilla. As it is known that the bone density is varied depends on the surgical sites, will it affect the results of GBR?

3. Only 12 month post operative date was provided and long term follow-up is recommended.

Author Response

We appreciate the time and effort each of the reviewer has dedicated to providing insightful feedback on ways to strengthen our paper. We have incorporated changes that reflect the suggestions you have graciously provided. To facilitate your review of our revision, the following is a point-by-point response to your questions and comments.

In this manuscript, the authors compared two low temperature-sintered ABBMs, Bio-Oss and A-Oss, which are used in GBR for the reconstruction of peri-implant dehiscence defect, in combination with the ultrafine titanium mesh. I would like to put forward the following questions and comments:

1. There are only 7 patients in each group, the sample size is comparatively small and may affects an RCT's reliability.

; Yes, we admit that the sample size seems small, however, it was determined to satisfy 90% power of the test at minimum at α = 0.05 according to the equal variance assumption based on the within-subject and between-subject means from the previous randomized controlled study comparing Bio-Oss and the calcium phosphate-coated anorganic bovine bone material in sinus augmentation [1]. Based on these results, the minimum sample size was estimated to be 9 subjects in each group considering a possible attrition rate of 10% during the study period [p 6, lines 198-204].

2. The implants and GBR in this experiment were placed in different sites, including both posterior and anterior areas on mandibular and maxilla. As it is known that the bone density is varied depends on the surgical sites, will it affect the results of GBR?

; Authors agree with the comment. Success of dental implants and GBR may depend on the density of the available bone in the recipient site, therefore, we adopted mixed-effect model to adjust the effects of covariates, such as gender, age, maxilla-mandible, and anterior-posterior, as a statistical analysis. 

3. Only 12 month post operative date was provided and long term follow-up is recommended.

; Very good point. We set postoperatively 12 months as the minimum period to evaluate the implant survival and the regenerated bone in our research timetable, however, we will try collecting more data in longer follow-ups.

Thank the reviewer again for giving us the opportunity to strengthen our manuscript with your valuable comments and queries. We hope that these revisions persuade you to accept our submission. Please stay safe and well during this COVID-19 pandemic.

Reference

1. Pang, K.-M.; Lee, J.-K.; Choi, S.-H.; Kim, Y.-K.; Kim, B.-J.; Lee, J.-H. Maxillary sinus augmentation with calcium phosphate double-coated anorganic bovine bone: Comparative multicenter randomized clinical trial with histological and radiographic evaluation. Implant dentistry 2019, 28, 39-45.

Reviewer 2 Report

1.Is the manuscript relevant and interesting?   The article is relevant and interesting.   2.How original is the topic?   The topic is current.   3.What does it add to the subject area compared with other published material?   The authors have collected and analyzed original data.   4. Is the paper well written?   Yes, the article is well written.   5. Is the text clear and easy to read?   Minor English editing is required.   6. Are the conclusions consistent with the evidence and arguments presented?   Yes, the conclusions consistent with the evidence and arguments presented.   7. Do they address the main question posed?  Yes, the Authors addressed the main question posed.   Other comments:   ·         English language: Minor English editing is required.   ·         Introduction: This section needs few improvements. For example, Authors may include a brief sentence at the beginning of this section regarding innovations in implant dentistry based on the following reference: <<Innovative materials and technologies to improve treatment outcomes, reducing at the same time morbidity, biological, and surgical times are an intense research topic in dentistry [https://doi.org/10.3390/jpm12010108]>>.  ·           ·         Materials and methods: This section has been properly prepared.  ·         Results: This section has been properly prepared.  ·         Discussion: What is the main theme that emerges from the authors' analysis? Is the study design a limitation? Please improve.  ·         Conclusion: This section has been properly prepared.  After making the indicated changes, the article may be suitable for publication after Editorial evaluation.  Thanks for the opportunity to review this manuscript.

Author Response

We appreciate the time and effort the reviewer has dedicated to providing insightful feedback on ways to strengthen our paper. We have incorporated changes that reflect the suggestions you have graciously provided. To facilitate your review of our revision, the following is a point-by-point response to your questions and comments.

1. Is the manuscript relevant and interesting? - The article is relevant and interesting. How original is the topic? - The topic is current. What does it add to the subject area compared with other published material? - The authors have collected and analyzed original data. Is the paper well written? - Yes, the article is well written. Is the text clear and easy to read? - Minor English editing is required. Are the conclusions consistent with the evidence and arguments presented? - Yes, the conclusions consistent with the evidence and arguments presented. Do they address the main question posed? - Yes, the Authors addressed the main question posed. 

; Thank you so much for encouraging remarks.

2. Other comments: English language: Minor English editing is required.

; I'm sorry that this manuscript undertook the native's language editing and the proof will be given if required.

3. Introduction: This section needs few improvements. For example, Authors may include a brief sentence at the beginning of this section regarding innovations in implant dentistry based on the following reference: <<Innovative materials and technologies to improve treatment outcomes, reducing at the same time morbidity, biological, and surgical times are an intense research topic in dentistry [https://doi.org/10.3390/jpm12010108].

; According to your suggestion, the sentence from that reference has replaced the one at the beginning of the Introduction section [p 2, lines 43-44]. 

4. Materials and methods: This section has been properly prepared. Results: This section has been properly prepared. 

; Thank you so much for encouraging remarks.

5. Discussion: What is the main theme that emerges from the authors' analysis? Is the study design a limitation? Please improve.          

; The main theme of our study is that the A-Oss (test group) could be comparable to Bio-Oss (control group) as an anoganic bovine bone material for guided bone regeneration (GBR) in implant dentistry. It has been stated in the Discussion section [p 10, lines 286-289]. The main limitation of our study is that bone quality was grossly assessed by the corticalization of regenerated bone using the periodontal probe. It has been added in the Discussion section, too [p 12, lines 355-357]. 

6. Conclusion: This section has been properly prepared. After making the indicated changes, the article may be suitable for publication after Editorial evaluation. Thanks for the opportunity to review this manuscript.

; Thank the reviewer again for giving us the opportunity to strengthen our manuscript with your valuable comments and queries. We hope that these revisions persuade you to accept our submission. Please stay safe and well during this COVID-19 pandemic.